# The Atypical Chemerin Receptor GPR1 Displays Different Modes of Interaction with β-Arrestins in Humans and Mice with Important Consequences on Subcellular Localization and Trafficking

**DOI:** 10.3390/cells11061037

**Published:** 2022-03-18

**Authors:** Gaetan-Nagim Degroot, Valentin Lepage, Marc Parmentier, Jean-Yves Springael

**Affiliations:** 1Institut de Recherche Interdisciplinaire en Biologie Humaine et Moléculaire (IRIBHM), Université Libre de Bruxelles (ULB), 1070 Brussels, Belgium; gaetan-nagim.degroot@ulb.be (G.-N.D.); valentin.lepage@ulb.be (V.L.); marc.parmentier@ulb.be (M.P.); 2Walloon Excellence in Life Sciences and Biotechnology (Welbio), 1300 Wavre, Belgium

**Keywords:** chemerin, ACKR, GPR1, β-arrestins, signaling

## Abstract

Atypical chemokine receptors (ACKRs) have emerged as a subfamily of chemokine receptors regulating the local bioavailability of their ligands through scavenging, concentration, or transport. The biological roles of ACKRs in human physiology and diseases are often studied by using transgenic mouse models. However, it is unknown whether mouse and human ACKRs share the same properties. In this study, we compared the properties of the human and mouse atypical chemerin receptor GPR1 and showed that they behave differently regarding their interaction with β-arrestins. Human hGPR1 interacts with β-arrestins as a result of chemerin stimulation, whereas its mouse orthologue mGPR1 displays a strong constitutive interaction with β-arrestins in basal conditions. The constitutive interaction of mGPR1 with β-arrestins is accompanied by a redistribution of the receptor from the plasma membrane to early and recycling endosomes. In addition, β-arrestins appear mandatory for the chemerin-induced internalization of mGPR1, whereas they are dispensable for the trafficking of hGPR1. However, mGPR1 scavenges chemerin and activates MAP kinases ERK1/2 similarly to hGPR1. Finally, we showed that the constitutive interaction of mGPR1 with β-arrestins required different structural constituents, including the receptor C-terminus and arginine 3.50 in the second intracellular loop. Altogether, our results show that sequence variations within cytosolic regions of GPR1 orthologues influence their ability to interact with β-arrestins, with important consequences on GPR1 subcellular distribution and trafficking.

## 1. Introduction

Atypical chemokine receptors (ACKRs) constitute a subgroup of chemokine receptors that do not induce G protein activation or cell migration [1,2]. Nevertheless, ACKRs play important biological functions in vivo by shaping the chemokines’ gradient or regulating the function of canonical chemokine receptors (CCKRs), making them interesting therapeutic targets in the context of inflammation and cancers [3,4,5,6]. Besides their role in the regulation of ligand availability, some ACKRs are also reported to trigger signaling through the recruitment of β-arrestins [7,8,9,10]. Interactions between GPCRs and β-arrestins were initially believed to provide a means to terminate G-protein signaling by preventing access to the G proteins. However, it was also demonstrated that β-arrestins can serve as scaffold proteins for signaling molecules such as ERK and c-Jun MAP kinases in order to trigger alternative signaling pathways [11,12,13]. Due to their higher propensity to activate β-arrestins than G proteins, ACKRs are often considered natural examples for β-arrestin-biased GPCRs, which makes them interesting models to study the concept of biased agonism [8,9]. The small subfamily of chemerin receptors are structurally and functionally related to chemokine receptors and is characterized by the fact that it comprises two atypical receptors, CCRL2 and GPR1, for one fully functional receptor, CMKLR1 [14]. Chemerin is a small 16 kDa protein structurally unrelated to chemokines and is involved in various pathophysiological processes, including inflammation, lipid, and glucose metabolism, angiogenesis, and cancer [15,16,17]. Chemerin is a chemoattractant factor for macrophages, myeloid and plasmacytoid dendritic cells (DCs), and natural killer (NK) cells, but has a role as an adipokine as well [16,18]. Chemerin binding to its canonical receptor CMKLR1 inhibits cAMP accumulation, induces intracellular calcium mobilization, triggers MAP kinase cascades, and recruits β-arrestins, which promote receptor internalization [14,19]. In contrast, chemerin binding to CCRL2 does not promote G protein or β-arrestin signaling, nor does it induce receptor internalization [14,20]. According to the current model, CCRL2 is an atypical receptor, devoid of signaling capacity but able to increase the local concentration of chemerin and to present the ligand to other cells expressing CMKLR1 [20]. CCRL2 was provisionally renamed ACKR5 pending the formal demonstration of its biological role [1]. Little is known regarding the third chemerin receptor, GPR1. Chemerin binding to GPR1 hardly activates any G protein but leads to efficient β-arrestin recruitment, receptor internalization, and chemerin scavenging [14,21,22]. It also triggers the phosphorylation of ERK1/2 MAP kinases, although to a much weaker extent than CMKLR1. Phosphorylation of ERK1/2 downstream of GPR1 requires β-arrestin 2 but not β-arrestin 1. However, it is also sensitive to *Pertussis toxin*, supporting a role of G_αi_ proteins in β-arrestin 2-dependent signaling [14]. G protein and β-arrestin signaling have for long been considered separable pathways; however, there is now a growing body of evidence that some level of coordination exists between the two pathways [23,24]. Thus, although not activating effectors downstream GPR1 in a conventional manner, G proteins may participate in some aspects of β-arrestin signaling. These properties make GPR1 a prototypical example of an atypical chemerin receptor naturally biased for β-arrestin. Although GPR1 shares many properties with atypical chemokine receptors ACKRs and should behave like them as a receptor shaping chemerin gradient, its biological role is still largely unknown. GPR1 KO mice were described to display a significant decrease in serum testosterone level, a lower bone mineral density, and glucose intolerance on a high-fat diet; however, how GPR1 and chemerin contribute to these alterations is unclear [25,26]. Thus, a better understanding of mouse GPR1 properties could help to appreciate its biological functions.

In this study, we compared the properties of human hGPR1 and mouse mGPR1 and found that they behave differently regarding their interaction with β-arrestins. hGPR1 interacts with β-arrestins as a result of chemerin stimulation, whereas its murine orthologue mGPR1 displays a strong constitutive interaction with β-arrestins in basal conditions. We investigated whether this behavior may influence other properties of mGPR1 and found that it is associated with an important localization of mGPR1 in early and recycling endosomes. We also found that chemerin induces the endocytosis of both receptors, but that the contribution of β-arrestins to this process is much more important for mGPR1 than for hGPR1. Nevertheless, both hGPR1 and mGPR1 scavenge chemerin and activate MAP kinases to the same extent. Finally, we found that arginine 3.50 in the ICL2 and the receptor C-terminus contribute to the constitutive interaction of mGPR1 with β-arrestins.

## 2. Material and Methods

### 2.1. Reagents, Plasmids, and Cell Lines

Recombinant human (aa21-157) and mouse (aa17-156) chemerin proteins and chemerin ELISA kits were purchased from BioTechne (Abingdon, UK). Plasmids encoding GPR1 and β-arrestin constructs were described elsewhere [14]. The Rluc and Venus tags are inserted, respectively at the N-terminus of arrestins and the C-terminus of all the h/mGPR1 constructs without the addition of any linker. Plasmids encoding β-arrestin1-Rluc is a gift from S. Marullo (Institut Cochin, Paris, France) [27]. Plasmid encoding GFP-ERK2 was a gift from R. Seger (Addgene plasmid # 37145) [28]. Membrane acceptors KRas-Venus, Rab5-Venus, Rab7-Venus, and Rab11-Venus were kindly provided by N. Lambert (Augusta University, USA) [29]. Chimeric h/m GPR1 receptors were generated by custom gene synthesis (GenScript) and cloned into pcDNA3(+)-C-RLuc or pcDNA3(+)-C-Venus. The HEK293 cell line lacking β-arrestins was provided by A. Inoue (Graduate School of Pharmaceutical Sciences, Tohoku University, Japan) [30]. HEK293 and HEK293T cells were cultured in Dulbecco’s modified Eagle’s medium supplemented with 10% fetal bovine serum (GIBCO), 100 U/mL penicillin, and 100 µg/mL streptomycin (Invitrogen). Cells were transiently transfected by using the calcium phosphate method as previously described [31].

### 2.2. β-arrestins BRET Assay

β-arrestins recruitment was measured by using a BRET proximity assay as previously described [30]. Briefly, plasmids encoding *R*Luc-β-arrestins and receptors fused to Venus were cotransfected into HEK293 or HEK293T cells. Then, 24 h post-transfection, cells were collected and seeded in 96-well microplates (165306, Nunc) and cultured for an additional 24 h. Cells were then incubated for at least two hours with 5 µM Enduren (Promega) before stimulation with 100 nM h or m chemerin. This concentration is above Kd (0.5–1 nM) and was successfully used to stimulate GPR1 in our previous studies [30]. The BRET^1^ signal between *R*Luc and Venus was measured at 25 °C to slow down the kinetics of β-arrestins recruitment and improve the temporal resolution. BRET readings were collected using an Infinite F200 reader (Tecan, Mechelen, BE). The BRET signal was calculated as the ratio of emission of Venus (520–570 nm) to *R*Luc (370–480 nm).

### 2.3. Subcellular Localization and Trafficking

Plasmids encoding receptors or β-arrestins fused to RLuc were cotransfected with either KRas-Venus, Rab5-Venus, Rab7-Venus, or Rab11-Venus. Then, 24 h post-transfection, cells were collected and seeded in 96-well microplates (165306, Nunc) and cultured for an additional 24 h. Cells were then incubated for at least two hours with 5 µM Enduren (Promega) before stimulation with 100 nM h or m chemerin. BRET^1^ signal between RLuc and Venus was measured at 37 °C to favor receptor internalization. BRET readings were collected using an Infinite F200 reader (Tecan). The BRET signal was calculated as the ratio of emission of Venus (520–570 nm) to RLuc (370–480 nm).

### 2.4. BRET Proximity Assay

BRET titration curves were obtained with HEK293T cells transfected with a constant amount of β-arrestin-*R*Luc and increasing amounts of receptors fused to Venus. BRET_Max_ values were determined by GraphPad Prism. Mock-transfected cells were used as a control in order to subtract raw basal luminescence and fluorescence from the data.

### 2.5. Chemerin Scavenging

Growth medium of CHO-K1 cells stably expressing hGPR1 or mGPR1 were stimulated with 25 nM h or m chemerin in Dulbecco’s modified Eagle’s medium supplemented with 1% bovine serum albumin for various times and chemerin present in the culture medium was quantified by ELISA. Mock-transfected cells were used as control.

### 2.6. MAP Kinase Assay

CHO-K1 cells stably expressing hGPR1 or mGPR1 were starved for 16 h in a serum-free medium prior to stimulation. Cells were stimulated with 50 nM h or m chemerin for various times, then collected by centrifugation and heated to 100 °C for 5 min in 2 × Laemmli sample buffer. For fractionation of membrane compartments, cells were suspended in a hypotonic buffer (10 mM HEPES pH 7.9 containing 10 mM KCl, 0.1 mM EDTA, 0.1 mM EGTA, 1 mM DTT, 0.5 mM PMSF, and a protease inhibitor cocktail (Roche)). Cells were lysed by adding 0.1 g/L Nonidet P-40 and vortexing vigorously. Nuclei were pelleted by centrifugation at 12,000× *g* for 30 min. Supernatants, corresponding to the cytosolic fractions, were saved for analysis. The nuclei were suspended in hypertonic buffer (20 mM HEPES pH7.9 containing 400 mM NaCl, 1 mM EDTA, 1 mM EGTA, 0.2 g/L glycerol, 1 mM DTT, 0.5 mM PMSF, and a protease inhibitor cocktail) and incubated for 30 min on a shaking platform at 4 °C. The samples were centrifuged at 14,000× *g* for 5 min and the supernatants, corresponding to the nuclear fractions, were saved. Whole-cell lysates and fractions were resolved on 10% Tris-Glycine polyacrylamide gels and transferred to nitrocellulose membranes. Phosphorylated ERK1/2 and total ERK1/2 were detected by using rabbit polyclonal anti-phospho-ERK1/2 (Cell Signaling, Danver, MA, USA #4370, 1:1000) and anti-ERK1/2 (Cell Signaling #4695S, 1:2000) antibodies. Chemiluminescent detection was performed using the ECL-Plus reagent (Perkin Elmer, Every, FR).

### 2.7. Statistical Analysis

Results are expressed as arithmetic means ± SEM. Significance was determined using one-way analysis of variance, followed by Tukey’s test (Prism6 software, GraphPad). For all tests, values of *p* < 0.05 were considered significant.

## 3. Results

### 3.1. mGPR1 Constitutively Interacts with β-Arrestins 1 and 2

We first compared the ability of mGPR1 and hGPR1 to recruit β-arrestins by using a BRET proximity assay. Low BRET signals are detected between the *Renilla* luciferase-tagged β-arrestins (β-arrestin-1-*R*Luc or β-arrestin-2-*R*Luc) and the Venus-tagged hGPR1. Upon chemerin stimulation, BRET signals increase gradually, reflecting the progressive interaction of β-arrestins with hGPR1. In striking contrast, much higher BRET signals are detected between β-arrestins-*R*Luc and the Venus-tagged mGPR1 in the basal condition (Figure 1A). Upon chemerin stimulation, the BRET signal increases slightly for β-arrestin 1 but is almost undetectable for β-arrestin 2, suggesting a high level of constitutivity. Moreover, the BRET variation detected for β-arrestin 1 is extremely fast (<1 min), supporting a conformational change in a preformed mGPR1/β-arrestin 1 complex rather than the recruitment of additional β-arrestin 1 molecules (Figure 1A). Similar results were obtained with the rat β-arrestin 2, which only contains a single T/I substitution at position 367 with respect to the mouse β-arrestin 2 (Appendix A). We also performed BRET titration experiments, in which cells were transfected with a fixed amount of β-arrestins-*R*Luc and increased amounts of Venus-tagged receptors. As shown in Figure 1B, BRET signals detected with mGPR1 are of much larger magnitude than those of hGPR1, thus confirming the strong constitutive interaction of mGPR1 with β-arrestins in the absence of chemerin stimulation.

### 3.2. mGPR1 Partially Localizes in Early and Recycling Endosomes

Next, we investigated whether the constitutive interaction of mGPR1 with β-arrestins alters the cell surface expression of the receptor. We relied on a BRET-based assay that detects the presence of the receptor in defined subcellular compartments by measuring the BRET signals between mGPR1-*R*Luc and the plasma membrane acceptor KRas-Venus, the early endosome acceptor Rab5a-Venus, the late endosome acceptor Rab7-Venus, or the recycling endosome acceptor Rab11-Venus. These BRET assays have proven to be very powerful tools to study the subcellular distribution and cellular trafficking of GPCRs in living cells and in real time. Compared with hGPR1, mGPR1 appears much less present at the plasma membrane and more present in early and recycling endosomes (Figure 2). A weak signal is also detected for both receptors in late endosomes. Upon chemerin stimulation, the BRET signals’ variation reveals the gradual removal of hGPR1 and mGPR1 from the plasma membrane and their relocation to early endosomes. The endocytosis of both receptors occurs with similar potencies, but the kinetics of the disappearance of mGPR1 from the plasma membrane appears faster than that of hGPR1 (K = 1.18 ± 0.12 vs. 0.38 ± 0.03, *p* = 0.003). Weak or no BRET signal changes are detected for recycling and late endosomes.

### 3.3. β-Arrestins Differentially Contribute to the Cellular Distribution and Trafficking of hGPR1 and mGPR1

We also tested whether β-arrestins are involved in the cellular distribution and trafficking of GPR1 by using Arr1/2 KO HEK293 cells from which the β- arrestin *Arrb1* and *Arrb2* genes have been mutated by CRISPR/Cas9 gene editing [29]. hGPR1 is present at the plasma membrane to the same extent whether the cells express β-arrestins or not (Figure 3). However, it should be noted that BRET signals between the *R*Luc-tagged receptors and KRas-Venus in parental HEK293 cells are of lower magnitude than those detected in HEK293T cells. This difference is likely due to the lower expression of the receptors in HEK293 cells lacking the large T antigen. Upon chemerin stimulation, hGPR1 is progressively relocated from the plasma membrane to early endosomes, although internalization appears slightly reduced in cells lacking β-arrestins. This result suggests that β-arrestins modestly contribute to the subcellular distribution and internalization of hGPR1. In contrast, mGPR1 appears much more present at the plasma membrane in cells lacking β-arrestins than in the parental cells. Nevertheless, parental and β-Arr1/2 KO cells have similar amounts of mGPR1 in early endosomes. Importantly, mGPR1 is totally refractory to chemerin-induced internalization in Arr1/2 KO cells. The contribution of β-arrestins to cellular distribution and trafficking thus appears completely different for mGPR1 and hGPR1.

### 3.4. Both hGPR1 and mGPR1 Scavenge Chemerin

We next investigated the ability of mGPR1 to scavenge chemerin and showed that it captures chemerin from the environment as efficiently as hGPR1 (Figure 4). As a control, we showed that the amount of chemerin remains almost constant in the supernatant of mock-transfected cells, ruling out any significant degradation of chemerin for the duration of the experiment.

### 3.5. Both hGPR1 and mGPR1 Activate MAP Kinases ERK1/2

We tested whether the constitutive interaction of mGPR1 with β-arrestins modifies the subcellular localization of β-arrestins by measuring the BRET signal between β-arrestins-RLuc and KRas-Venus. In cells expressing mGPR1, β-arrestins partially localize to the plasma membrane in basal conditions (Figure 5). Chemerin stimulation further increases the BRET signals, supporting additional translocation of new β-arrestin molecules and/or a conformation change within preformed mGPR1/β-arrestin complexes. By comparison, in cells expressing hGPR1, β-arrestins shows no or weak localization at the plasma membrane in basal conditions compared to the situation after chemerin stimulation. We also showed that the constitutive interaction of mGPR1 with β-arrestins brings ERK2 in close proximity of mGPR1 in basal conditions. Chemerin stimulation does not further increase the BRET signal, suggesting no or weak recruitment of additional β-arrestin/ERK2 complexes. By comparison, the BRET signal between hGPR1 and ERK2 is very low in basal conditions and chemerin stimulation slightly increases the BRET signal, reflecting the gradual increase in proximity between GPR1 and ERK2. We next wonder whether this pre-assembly of a mGPR1/β-arrestin/MAP kinase complex in basal conditions impacts the activation of the MAP kinases ERK1/2. mGPR1 triggers the activation of ERK1/2 to the same extent and with the same kinetics as hGPR1 (Figure 6A), indicating that mGPR1 stimulation is still mandatory to activate the β-arrestin-associated MAP kinases. One reported consequence of the formation of β-arrestin-ERK complexes is also the cytosolic retention of β-arrestin-bound ERK1/2 [32,33]. Fractionation studies reveal that hGPR1 and mGPR1 trigger the activation of a predominantly cytosolic pool of ERK1/2 (Figure 6B).

### 3.6. The Constitutive Interaction of mGPR1 with β-Arrestins Involves the Receptor C-Terminus and R3.50

Finally, we investigated the molecular basis underlying the constitutive interaction of mGPR1 with β-arrestins. It is well-documented that β-arrestins interact with GPCRs by using the C-terminus and intracellular loops (ICLs) of the receptors. Sequence alignment shows that hGPR1 and mGPR1 share 80% of sequence identity and 91% of sequence homology over their entire length and that few substitutions take place within their ICLs and the C-terminus (Figure 7). Analysis with the NetPhos 3.1 prediction server revealed that these regions of mGPR1 contain additional putative phosphorylation sites that may favor the interaction with β-arrestins (Figure 7). It is also well known that mGPR1 contains an arginine residue at position 3.50, whereas this position is occupied by a histidine in hGPR1 [34]. This substitution takes place within the highly conserved “DRY” motif involved in GPCR activation and G protein interaction. Naturally occurring or engineered mutations of R^3.50^ often impair GPCR signaling by decreasing their ability to couple to G proteins, but it was also reported that R^3.50^ mutations may favor the interaction of some GPCR with β-arrestins [35,36]. Thus, we generated two chimeric hGPR1 receptors, one in which the histidine residue at position 3.50 is replaced by an arginine (hGPR1-DRY) and a second in which the entire C-terminus is replaced by the C-terminus of mGPR1 (hGPR1-mCT), and tested their interaction with β-arrestins (Figure 8A,B, Appendix A). Replacing the C-terminus of hGPR1 by that of mGPR1 significantly increases the interaction of the receptor with β-arrestins 1 and 2 in basal conditions. Replacing the histidine at position 3.50 by an arginine also increases this interaction, although to a lower extent. Nevertheless, for both chimeric receptors, the extent of the constitutive interaction with β-arrestins remains lower compared to the situation encountered with mGPR1. Chemerin stimulation of the chimeric receptors further increases the interaction with β-arrestins, confirming their intermediate level of constitutivity (Figure 8C,D). We also showed that the distribution of these chimeric receptors between the plasma membrane and early endosomes is modified (Figure 9). The chimeric hGPR1-DRY is less abundant in the plasma membrane while its localization in endosomes is more important. This redistribution is even more drastic for the chimeric hGPR1-mCT, which is almost absent from the plasma membrane and essentially localized in early endosomes. Chemerin stimulation does not modify much this status, suggesting that hGPR1-mCT is almost completely relocalized to endosomes and thereby refractory to further endocytosis. These results confirm the importance of the constitutive interaction with β-arrestins for the subcellular localization of the receptor and show that sequence variation between GPR1 orthologs may also alter their trafficking properties.

## 4. Discussion

Atypical chemokine receptors (ACKRs) have emerged over the past years as key regulators of the chemokine network. However, a better understanding of their properties is still necessary to fully apprehend their biological roles in pathophysiological conditions. In this study, we focused on the functional characterization of the chemerin receptor GPR1, which shares many properties with ACKRs but has received little attention so far. We compared the properties of the human and mouse orthologs of GPR1, and it was revealed that they behave differently regarding their interaction with β-arrestins. Human hGPR1 recruits both β-arrestin 1 and 2 following ligand stimulation, whereas mouse mGPR1 interacts strongly with β-arrestins in basal conditions (Figure 10). Chemerin stimulation does not further increase the interaction of mGPR1 with β-arrestins, suggesting a high level of constitutivity. It should be noted that our results were obtained with human β-arrestin1/2, as well as with rat β-arrestin 2, making the hypothesis of an artifactual interaction of mGPR1 with β-arrestins unlikely. Unfortunately, we were not able to reach sufficient expression levels of β-arrestins and GPR1 in mouse cell lines to measure a BRET signal and rule out any influence of the cellular background, nor were we able to exclude the possibility that the expression of GPR1 in recombinant cells may not reflect its behavior in native cells. However, testing the interaction of GPR1 with β-arrestins in in vivo settings is extremely difficult because expression levels of GPR1 are very low, and there is currently no good antibody targeting mGPR1.

We argue that the constitutive interaction of mGPR1 with β-arrestins favors the presence of the receptor in early and recycling endosomes in basal conditions and the downmodulation of the receptor upon chemerin stimulation. In comparison, human hGPR1 appears to be more present at the plasma membrane and less in endosomal compartments, compared with mGPR1, and its subcellular localization and trafficking barely depend on the presence of β-arrestins (Figure 10). Due to the different properties of human and mouse GPR1, it is difficult to link the function of this receptor to those of other known ACKRs. mGPR1 seems to behave similarly to ACKR2-4, which interact to varying degrees with β-arrestins in basal conditions and localize preferentially in endosomes [5,31,32,33,34]. However, β-arrestins seem dispensable for ACKR2-4 internalization, whereas they are mandatory for the chemerin-induced internalization of mGPR1. β-arrestins are also dispensable for the internalization of human hGPR1, which barely interacts with β-arrestins in basal conditions. Thus, the subcellular distribution and trafficking of GPR1 appear to vary between species due to different modes of interaction with β-arrestins. It is also possible that the internalization of GPR1 occurs through both β-arrestin-dependent and -independent mechanisms according to the receptor expression site or environmental conditions, and that circumstances favoring receptor pre-coupling with β-arrestins prevent the activation of β-arrestin-independent mechanisms. It is interesting to note that human ACKR4, which displays an intermediate level of constitutivity toward β-arrestin in basal conditions, is internalized via β-arrestin-dependent and -independent mechanisms [37]. The constitutive interaction of mGPR1 with β-arrestins alters neither the ability of mGPR1 to scavenge chemerin from the environment nor the downstream signaling of the receptor. We previously showed that the activation of MAP kinases ERK1/2 by human hGPR1 requires both β-arrestin 2 and Gαi proteins [14]. However, it is unlikely that the Gαi proteins play a direct role in this process, as our fractionation studies reveal that the pool of activated ERK1/2 is mostly cytosolic. Our results are rather in favor of the role of Gαi proteins in the activation of the β-arrestin-bound pool of ERK1/2. Whether mGPR1 interacts constitutively with β-arrestins does not seem to impact the activation of ERK1/2. However, we cannot exclude formally that it may influence the activation of other β-arrestin-bound molecules.

In this study, we also explored the molecular basis underlying the constitutive interaction of β-arrestins with mGPR1. Using chimeric h/m GPR1, we showed that the C-terminus of mGPR1 is involved in its basal interaction with β-arrestins. The presence of additional phosphorylation sites in the C-terminus of mGPR1 might explain its greater propensity to interact with β-arrestins. Our results are thus in line with several other studies reporting the importance of GPCR C-termini in the interaction with β-arrestins and with the “barcoding hypothesis” proposing that a phosphorylation pattern regulates the interaction of GPCRs with β-arrestins [37,38,39,40,41]. We also showed in this study that the replacement of histidine 3.50 of hGPR1 by an arginine is sufficient to increase the basal interaction of hGPR1 with β-arrestins, and to promote a partial redistribution of the receptor from the plasma membrane to early endosomes. This result confirms that, besides the C-terminus, GPR1 ICLs also participate in the interaction with β-arrestins [42]. Alignment of all available GPR1 sequences revealed the presence of a histidine residue at position 3.50 in primates, whereas all other species share an arginine. Whether the histidine in these receptors also reduces their basal interaction with β-arrestins is currently unknown. Altogether, our results confirm that multiple determinants are required for the basal interaction of mGPR1 with β-arrestins and that the substitution of a single residue can influence the receptor localization, trafficking, and signaling.

The biological functions of the atypical receptor GPR1 have not yet been fully apprehended. Several studies aimed to tackle this issue by using mice invalidated for GPR1. However, our data reveal that the properties of GPR1 in mice might not exactly reflect its behavior in humans due to sequence variations in the C-terminus of the receptor and the differences in their interactions with β-arrestins. Closer examination of β-arrestin interactions in orthologs of other ACKRs or GPCRs could reveal similar variations in their ability to interact with β-arrestins, with important consequences on the functions of these receptors and the way to apprehend these functions in animal models.

## Figures and Tables

**Figure 1 cells-11-01037-f001:**
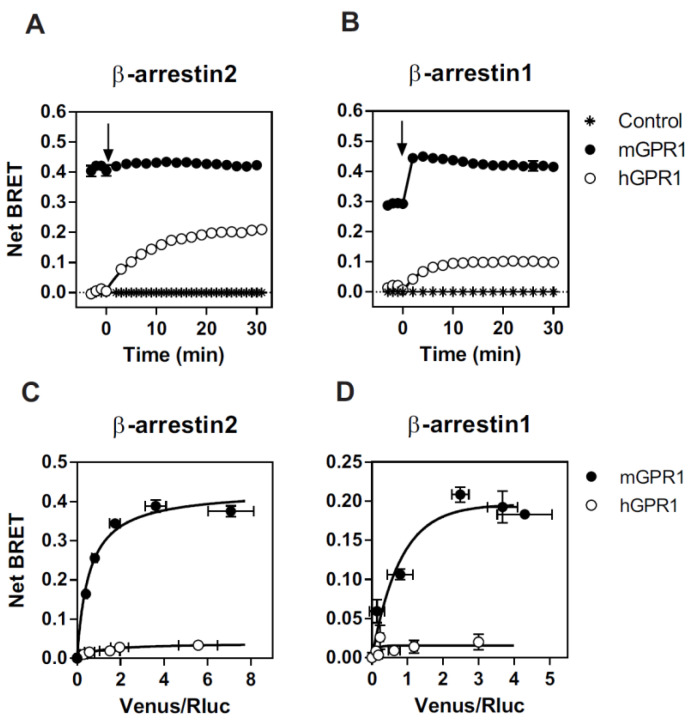
Mouse GPR1 constitutively interacts with β-arrestins: (**A**,**B**) real-time measurement of BRET signal in HEK293T cells expressing β-arrestin2-*R*Luc (**A**) or β-arrestin1-*R*Luc (**B**) in combination with hGPR1-Venus (○) or mGPR1-Venus (●), in basal conditions and after stimulation with 100 nM chemerin. (**C**,**D**) BRET titration curves obtained with HEK293T cells transfected with a constant amount of β-arrestin2-*R*Luc (**C**) or β-arrestin1-*R*Luc (**D**) and increasing amounts of hGPR1-Venus (○) or mGPR1-Venus (●). Results are expressed as Net BRET corresponding to the difference between the BRET signal measured between the donor and the acceptor pair and the BRET signal measured with the donor only. Data represent the mean ± SEM of at least three independent experiments.

**Figure 2 cells-11-01037-f002:**
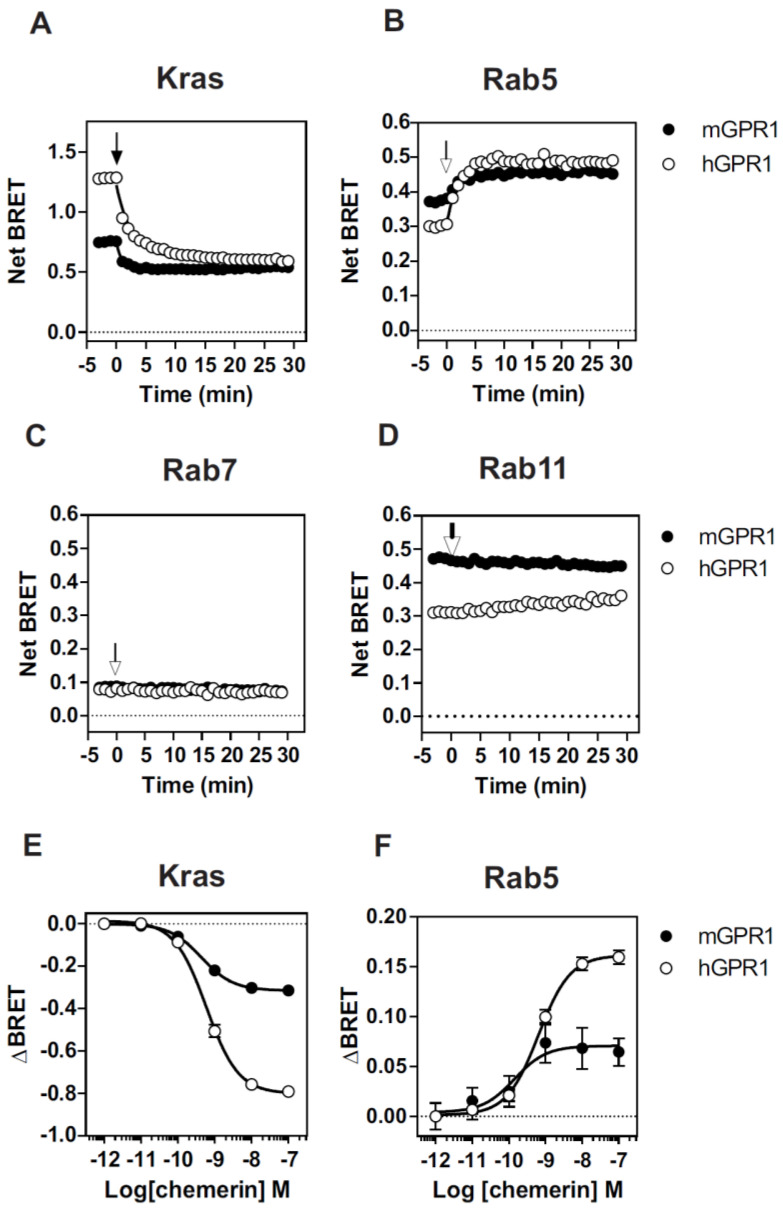
Cellular distribution and internalization of GPR1: (**A**–**D**) real-time measurement of BRET signal in HEK293T cells expressing hGPR1-*R*Luc (○) or mGPR1-*R*Luc (●) in combination with KRas-Venus (plasma membrane) (**A**), Rab5-Venus (early endosomes) (**B**), Rab7-Venus (late endosomes) (**C**) or Rab11-Venus (recycling endosomes) (**D**) in basal conditions and after stimulation with 100 nM chemerin. Results are expressed as Net BRET corresponding to the BRET signal measured between the donor and the acceptor minus the BRET signal measured with the donor only; (**E**,**F**) real-time measurement of BRET signal in cells expressing hGPR1-*R*Luc (○) or mGPR1-*R*Luc (●) in combination with KRas-Venus (**E**) or Rab5-Venus (**F**) measured 30 min after the simulation with increasing concentrations of chemerin. Results are expressed as ΔBRET corresponding to the difference between the BRET signal measured before and after stimulation with chemerin. Data represent the mean ± SEM of at least three independent experiments.

**Figure 3 cells-11-01037-f003:**
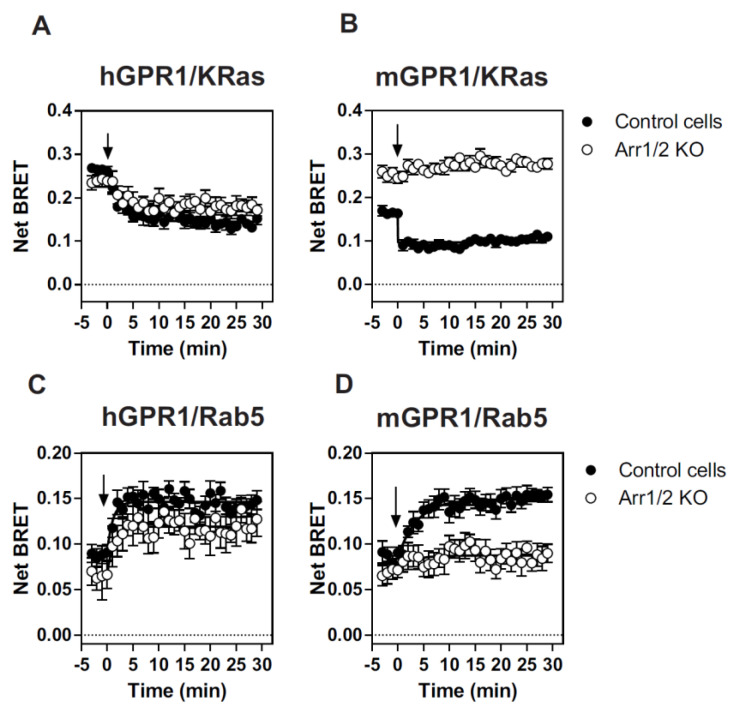
β-arrestins are involved in mGPR1 subcellular localization and trafficking. (**A**–**D**) Real-time measurement of BRET signal in WT (●) or β-arrestin 1/2 KO (○) HEK293 cells expressing hGPR1- RLuc or mGPR1-RLuc in combination with KRas-Venus (plasma membrane) (**A**,**B**) or Rab5-Venus (early endosomes) (**C**,**D**), in cells expressing or not basal conditions and after stimulation with 100 nM chemerin. Results are expressed as Net BRET corresponding to the BRET signal measured between the donor and the acceptor minus the BRET signal measured with the donor only. Data represent the mean ± SEM of at least three independent experiments.

**Figure 4 cells-11-01037-f004:**
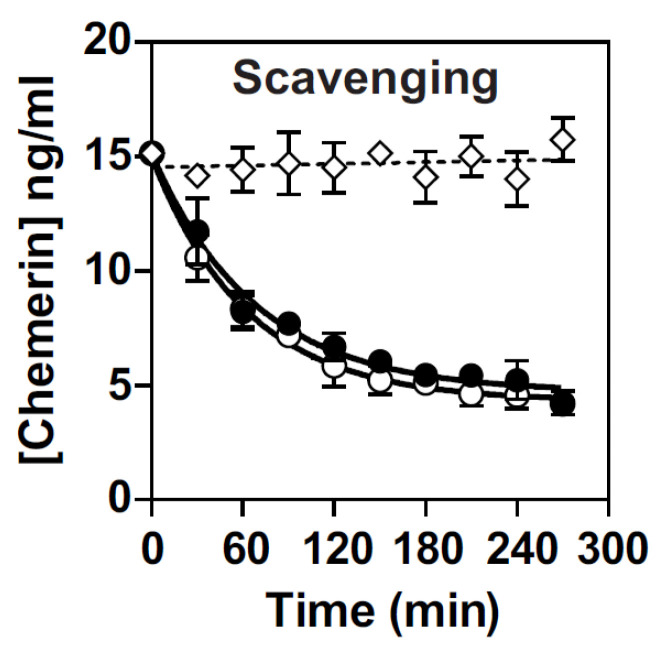
hGPR1 and mGPR1 scavenge chemerin similarly. Mock transfected CHO-K1 cells (◇) or cells stably expressing hGPR1-*R*Luc (○) or mGPR1-*R*Luc (●) were incubated with 25 nM chemerin for various times and the amount of chemerin remaining in the medium quantified by ELISA. Data represent the mean ± SEM of at least three independent experiments.

**Figure 5 cells-11-01037-f005:**
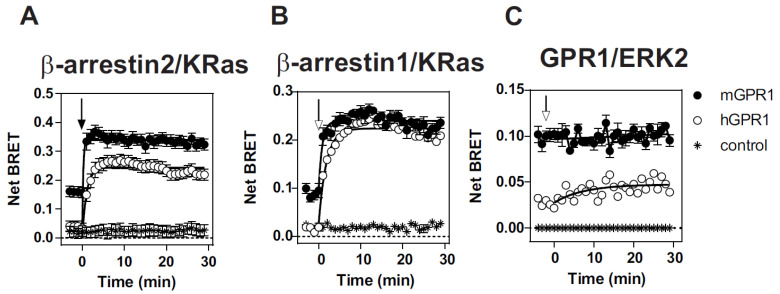
β-arrestins partially relocalize to the plasma membrane in cells expressing mGPR1. (**A**,**B**) Real-time measurement of BRET signal in HEK293T cells expressing β-arrestin2-*R*Luc (**A**) or β-arrestin1-*R*Luc (**B**) in combination with the plasma membrane acceptor KRas-Venus and hGPR1 (○) or mGPR1 (●), in basal conditions and after stimulation with 100 nM chemerin. Control curves (✳) correspond to cells transfected with β-arrestins fused to Rluc and KRas-Venus only. Results are expressed as Net BRET corresponding to the BRET signal measured between the donor and the acceptor minus the BRET signal measured with the donor only. Data represent the mean ± SEM of at least three independent experiments. (**C**). Real-time measurement of BRET signal in HEK293T cells expressing hGPR1-RLuc (○) or mGPR1-*R*Luc (●) in combination with ERK2-EYFP, in basal conditions and after stimulation with 100 nM chemerin. Control curves (✱) correspond to cells transfected with receptors fused to Rluc only. Data represent the mean ± SEM of at least three independent experiments.

**Figure 6 cells-11-01037-f006:**
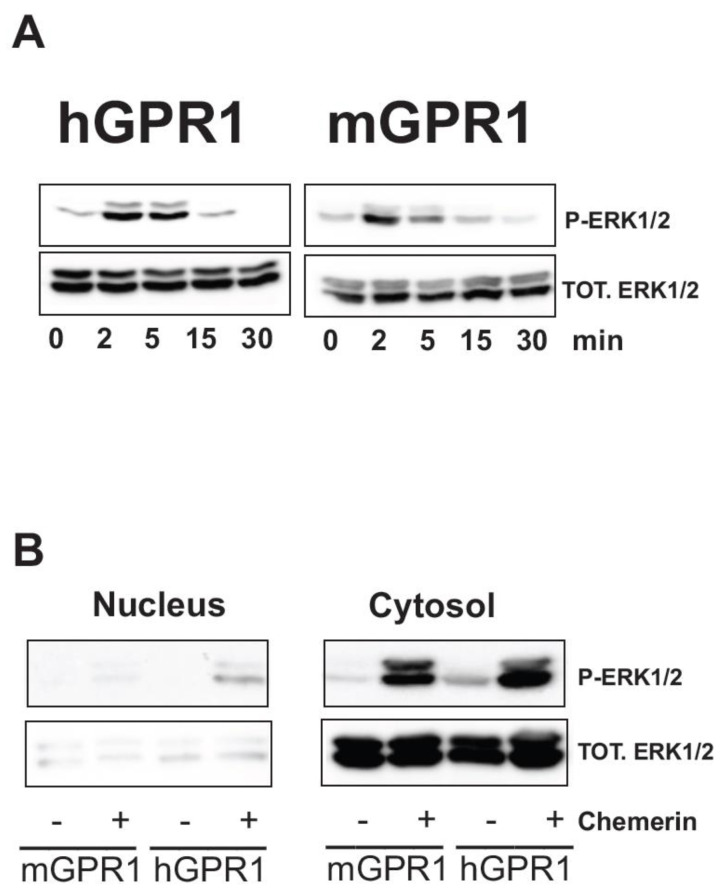
hGPR1 and mGPR1 activate MAP kinases ERK1/2. (**A**,**B**) Serum-starved CHO-K1 cells expressing hGPR1 or mGPR1 were stimulated with 50 nM chemerin for indicated times and the phospho-ERK1/2 content was determined by immunoblotting. The phospho-ERK1/2 content was analyzed in whole cell lysates (**A**) and in nuclear and cytosolic fractions (**B**). Detection of total ERK1/2 (lower panel) was used to ascertain that an equal amount of material was loaded in each lane. Quantitative data analysis was performed by using the ImageJ software. Data represent the mean ± SEM of three independent experiments.

**Figure 7 cells-11-01037-f007:**
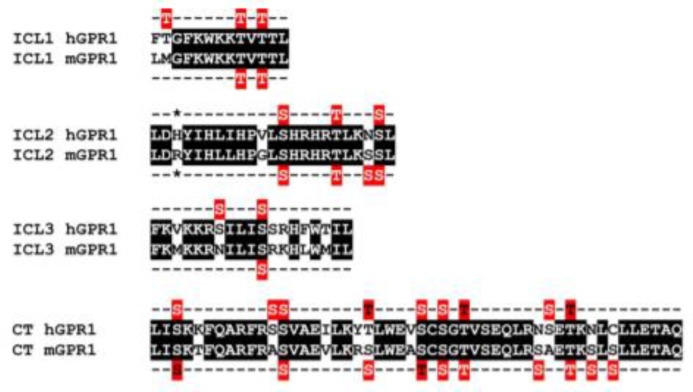
Sequence alignment of the ICLs and C-terminus of hGPR1 and mGPR1. Identical residues are shaded in black and S/T phosphorylation sites predicted by the NetPhos 3.1 software are highlighted in red. The 3.50 R/H residue in ICL2 is marked with a star.

**Figure 8 cells-11-01037-f008:**
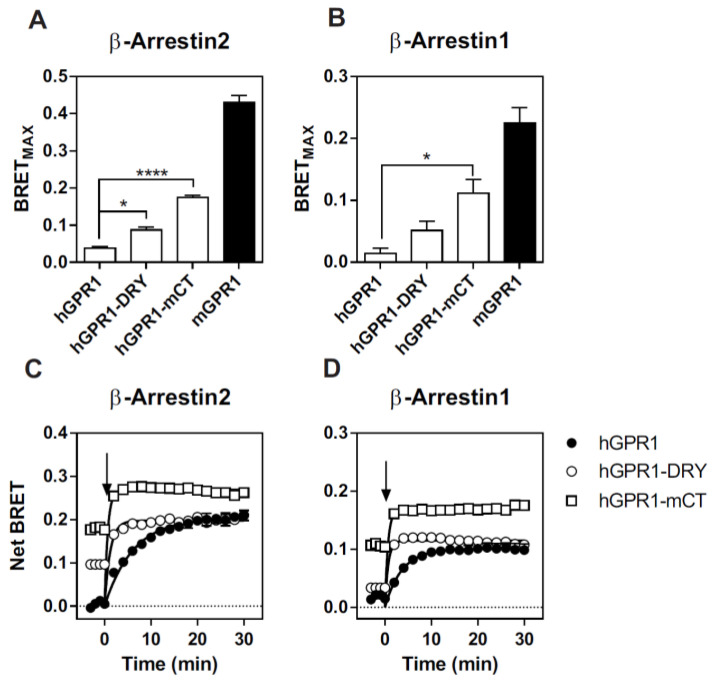
R^3.50^ and the C-terminus of mGPR1 are involved in its interaction with β-arrestins. (**A**,**B**) BRET_MAX_ values derived from BRET titration curves obtained with HEK293T cells transfected with a constant amount of β-arrestin2-*R*Luc (**A**) or β-arrestin1-*R*Luc (**B**) and increasing amounts of hGPR1-Venus, hGPR1-DRY-Venus, hGPR1-mCT or mGPR1-Venus. (**C**,**D**) Real-time measurement of BRET signal in HEK293T cells expressing β-arrestin2-*R*Luc (**C**) or β-arrestin1-*R*Luc (**D**) in combination with hGPR1-Venus (○), hGPR1-DRY-Venus (■) or hGPR1-mCT-Venus (□), in basal conditions and after stimulation with 100 nM chemerin. Results are expressed as Net BRET corresponding to the BRET signal measured between the donor and the acceptor minus the BRET signal measured with the donor only. Data represent the mean ± SEM of at least three independent experiments. * *p* < 0.05; **** *p* < 0.0001.

**Figure 9 cells-11-01037-f009:**
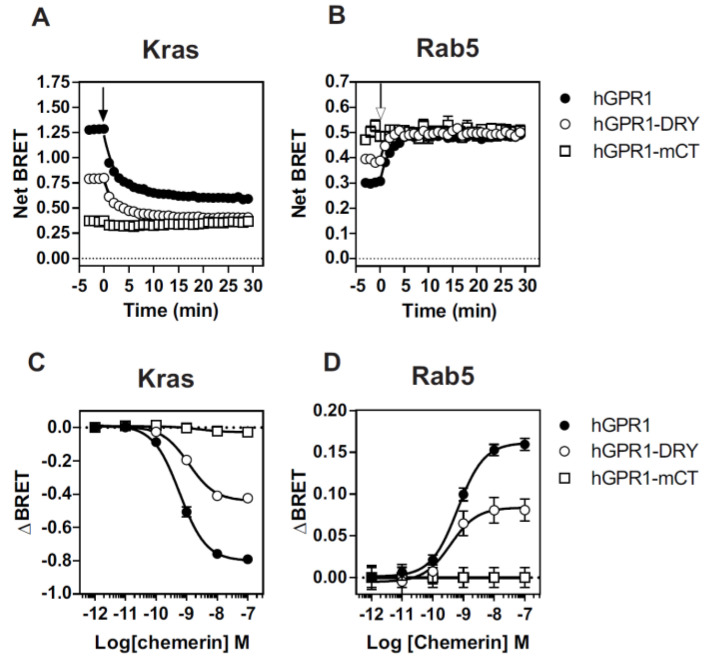
R^3.50^ and the C-terminus of mGPR1 are involved in its subcellular localization and trafficking. (**A**,**B**) Real-time measurement of BRET signal in HEK293T cells expressing hGPR1-RLuc (●), hGPR1-DRY-*R*Luc (○) or hGPR1-mCT-*R*Luc (□) in combination with the plasma membrane acceptor KRas-Venus (**A**) or the early endosome acceptor Rab5-Venus (**B**), in basal conditions and after stimulation with 100 nM chemerin. Results are expressed as Net BRET corresponding to the BRET signal measured between the donor and the acceptor minus the BRET signal measured with the donor only. (**C**,**D**) Real-time measurement of the BRET signal measured 30 min after simulation with increasing concentrations of chemerin. Results are expressed as ΔBRET corresponding to the difference between the BRET signal measured before and after stimulation with chemerin. Data represent the mean ± SEM of three independent experiments.

**Figure 10 cells-11-01037-f010:**
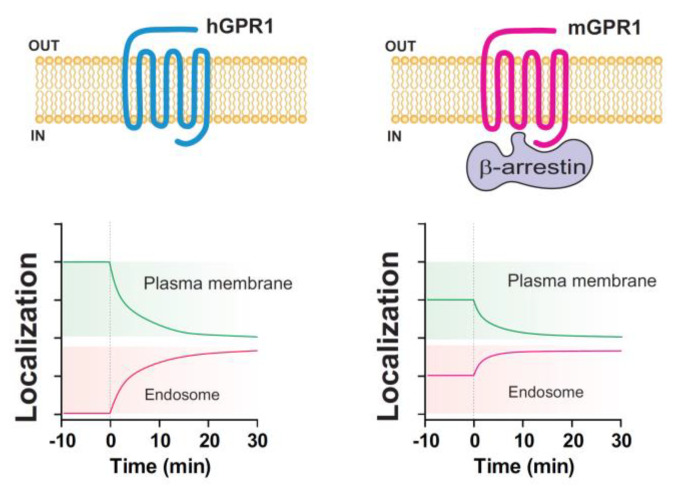
Overview of the main properties of human and mouse GPR1. In basal conditions, human hGPR1 interacts weakly with β-arrestins, whereas its mouse orthologue mGPR1 displays a strong constitutive interaction with β-arrestins. Constitutive interaction of mGPR1 with β-arrestins required different structural constituents, including the receptor C-terminus and arginine 3.50 in the second intracellular loop. hGPR1 is more present at the plasma membrane and less in endosomal compartments, compared with mGPR1. Thus, constitutive interaction of mGPR1 with β-arrestins favors the presence of the receptor in early and recycling endosomes in basal conditions. Both hGPR1 and mGPR1 are progressively relocated from the plasma membrane to early endosomes after chemerin stimulation (t = 0).

## Data Availability

The data that support the findings of this study are available from the corresponding author upon reasonable request.

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
