# Peer review of "The Atypical Chemerin Receptor GPR1 Displays Different Modes of Interaction with β-Arrestins in Humans and Mice with Important Consequences on Subcellular Localization and Trafficking"

_cells, 2022, doi:10.3390/cells11061037_

Round 1

Reviewer 1 Report

In their study Degroot et al. investigated the atypical chemerin receptor GPR1 and proposed that human and mouse receptor display different modes of interaction with β-arrestins leading to important consequences on the receptor localization and trafficking.

The question addressed in this study is interesting and results useful for the community as the comparative studies of receptors from different species are often neglected, especially in the field of cytokines where observations made in mice are too often directly transposed to human without critical consideration. 

Major comments

The authors present GPR1 as an atypical chemokine receptor. According to my knowledge, chemerin, although it shares some properties with chemokines is not one. Likewise, the authors present GPR1 as an atypical chemokine receptor but they indicate in lines 60 and 65 that GPR1 signalling is sensitive to pertussis toxin, indicative of G protein mediated signalling. However, the inclusion criteria for the ACKRs is the absence of G protein signalling and of migration following stimulation (lines 31 to 32). This aspect should be clarified and comparison between GPR1 and ACKR receptors should be made carefully.

My main concern regarding this study is the cellular background and effectors used to compare the mouse and human receptors. It is not clear to me which beta-arrestins were used, (human, mouse, rat?). Could the difference be due to the fact that the authors studied a mouse receptor in a human cell line with human arrestins? Would the results be the same in a mouse cell background? These aspects should be addressed with additional key experiments or at least discussed. In light of the statement in line 13-14 regarding the use of transgenic mouse models, I would also have expected the authors to compare hGPR1 in human and mouse cellular background. This could also be discussed.

A schematic summary of the human vs mouse GPR1 results would be helpful for the reader.

Material and methods sections is not complete

  • Section describing the scavenging assay described in section 3.4 and in figure 4 is missing.
  • The BRET ERK1/2-eYFP biosensor used in figure 5 is not described in M&M section
  • The title of section 2.2 should be modified. The second part (lines 96-97) of the title should be part of the first paragraph.
  • Please prove additional details regarding the engineered BRET fusions (linker, tag localisation).

Results

  • Text from lines 200 to 208 appears not related to MAP kinases activation part of the section 3.5 but rather to section 3.3 on receptor trafficking
  • Is there any reason why MAP kinase experiments were conducted in another cellular background (CHO-1 instead of HEK cells)?

Discussion

  • Are the differences in the properties of GPR1 orthologues also observed with other chemerin receptors, is there anything know? Likewise, is there any other similar observation for other cytokine receptors?

Figures

  • Identification of the panels A, B, C is missing in figures 1, 2, 3, 8, 9.

References

  • There is a problem of formatting for ref 1

Author Response

In their study Degroot et al. investigated the atypical chemerin receptor GPR1 and proposed that human and mouse receptor display different modes of interaction with β-arrestins leading to important consequences on the receptor localization and trafficking.

The question addressed in this study is interesting and results useful for the community as the comparative studies of receptors from different species are often neglected, especially in the field of cytokines where observations made in mice are too often directly transposed to human without critical consideration. 

Major comments

The authors present GPR1 as an atypical chemokine receptor. According to my knowledge, chemerin, although it shares some properties with chemokines is not one.

We thank the reviewer for this comment and modify the text to avoid any misunderstanding. We now refer GPR1 in the revised version of the manuscript as an “atypical chemerin receptor sharing properties with atypical chemokine receptors (ACKR)”.

Likewise, the authors present GPR1 as an atypical chemokine receptor but they indicate in lines 60 and 65 that GPR1 signalling is sensitive to pertussis toxin, indicative of G protein mediated signalling. However, the inclusion criteria for the ACKRs is the absence of G protein signalling and of migration following stimulation (lines 31 to 32). This aspect should be clarified and comparison between GPR1 and ACKR receptors should be made carefully.

We agree that these data may appear paradoxical. Nevertheless, recent studies clearly demonstrate that G-protein is necessary for certain aspects of β-arrestin signaling. Therefore, G-protein may also be important for the signaling of receptors naturally biased toward arrestin such as GPR1 and ACKRs. Introduction section has been modified in order to better explain this concept (lines 65-75).

My main concern regarding this study is the cellular background and effectors used to compare the mouse and human receptors. It is not clear to me which beta-arrestins were used, (human, mouse, rat?). Could the difference be due to the fact that the authors studied a mouse receptor in a human cell line with human arrestins? Would the results be the same in a mouse cell background? In light of the statement in line 13-14 regarding the use of transgenic mouse models, I would also have expected the authors to compare hGPR1 in human and mouse cellular background. This could also be discussed.These aspects should be addressed with additional key experiments or at least discussed.

We thank the reviewer for the critical comment regarding our BRET experiments. All experiments were performed with human arrestins and in human HEK293 cells. Although we did not test the interaction of mGPR1 with mouse arrestins, we do not believe that the constitutive interaction of mGPR1 with human arrestin is artifactual. Indeed, human and mouse arrestins share more than 98% of sequence identity. Moreover, we obtained similar results with the rat beta-arrestin 2 which bearing a single I/T substitution at position 367 regarding to the sequence of mouse mouse beta-arrestin 2. We also try to performed additional BRET experiments with mouse NIH3T3 and MEF cells; however, these cells are hard to transfect and we reached only 15-20% of transfection efficiency. Under these conditions, we were unable to detect a BRET signal between Rluc-Arrestin and GPR1-Venus. Testing the interaction of GPR1 with β-arrestins in in-vivo settings is also difficult because expression levels of GPR1 are very low and there is currently no good antibody targeting mGPR1. All these points are now discussed in the revised manuscript (lines292-300).

A schematic summary of the human vs mouse GPR1 results would be helpful for the reader.

We tried to summarize the human vs mouse GPR1 important results in a drawing. We leave it to the discretion of the editor to use this diagram as an addionnal figures (figure 10) or as graphical abstract.

 Material and methods sections is not complete

  • Section describing the scavenging assay described in section 3.4 and in figure 4 is missing.

A section describing the scavenging assay has been added.

  • The BRET ERK1/2-eYFP biosensor used in figure 5 is not described in M&M section

The reference of the biosensor has been added in section 1.1.

  • The title of section 2.2 should be modified. The second part (lines 96-97) of the title should be part of the first paragraph.

The title of section 2,2 has been corrected.

  • Please prove additional details regarding the engineered BRET fusions (linker, tag localisation).

Details regarding the BRET constructs have been added in section 1.1.

Results

  • Text from lines 200 to 208 appears not related to MAP kinases activation part of the section 3.5 but rather to section 3.3 on receptor trafficking.

This part of the text refers to the localization of arrestins which scaffold the MAP kinases and is also related to section 3.3.

Is there any reason why MAP kinase experiments were conducted in another cellular background (CHO-1 instead of HEK cells)?

We performed MAP kinase experiments in CHO-K1 cells stably expressing h/mGPR1 in order to improve the expression level of the receptors.

Discussion

  • Are the differences in the properties of GPR1 orthologues also observed with other chemerin receptors, is there anything know? Likewise, is there any other similar observation for other cytokine receptors?

We did not test other chemerin receptors or other GPCR in this study.

Figures

  • Identification of the panels A, B, C is missing in figures 1, 2, 3, 8, 9.

Identification of panels have been added.

References

  • There is a problem of formatting for ref 1

Format of Ref 1 has been corrected.

Reviewer 2 Report

The study describes  and  compares  properties of the human and mouse chemerin receptor GPR1, including their interaction with β-arrestins. The research is based on a series of interesting, complementary and well-planned experiments.

Minor concerns

  1. Line 86: please provide more details about the recombinant chemerin used.
  2. Subtitle 2.2: please modify the name of the subtitle; references are hardly ever cited in subtitle.
  3. In the Materials and methods section, the authors do not provide the concentration of chemerin, nor do they provide the basis on which they selected the concentration used.
  4. Lines 124 and 128: instead of 12,000 / 14,000 RPM, please provide values in g.
  5. Subsections 3.4 and 3.6: the way to obtain the results presented here was not described in the Materials and methods section.
  6. Figures 1, 2, 3, 5, 8 and 9: lack of letter markings (A-D) indicated in figure captions.
  7. Lines 349-350: meaning of the symbols “○, ●”are different in figure 3 and in the figure caption. 
  8. Line 358: “50 pmoles” - what does it mean?
  9. The symbols used in Figure 9 and the caption for this figure are different.

Author Response

The study describes  and  compares  properties of the human and mouse chemerin receptor GPR1, including their interaction with β-arrestins. The research is based on a series of interesting, complementary and well-planned experiments.

Minor concerns

  1. Line 86: please provide more details about the recombinant chemerin used.
  2. Subtitle 2.2: please modify the name of the subtitle; references are hardly ever cited in subtitle.
  3. In the Materials and methods section, the authors do not provide the concentration of chemerin, nor do they provide the basis on which they selected the concentration used.
  4. Lines 124 and 128: instead of 12,000 / 14,000 RPM, please provide values in g.
  5. Subsections 3.4 and 3.6: the way to obtain the results presented here was not described in the Materials and methods section.
  6. Figures 1, 2, 3, 5, 8 and 9: lack of letter markings (A-D) indicated in figure captions.
  7. Lines 349-350: meaning of the symbols “○, ●”are different in figure 3 and in the figure caption. 
  8. Line 358: “50 pmoles” - what does it mean?
  9. The symbols used in Figure 9 and the caption for this figure are different.

We thank this reviewer for the critical reading of the manuscript and his comments. Text and figures have been modified in the revised manuscript in order to correct errors and omissions.